# Validity Study for Clinical Use of Hand-Held Spirometer in Patients with Chronic Obstructive Pulmonary Disease

**DOI:** 10.3390/healthcare12050507

**Published:** 2024-02-20

**Authors:** Byeong-Soo Kim, Sam-Ho Park, Sung-Soo Jung, Hong-Jun Kim, Seong-Dae Woo, Myung-Mo Lee

**Affiliations:** 1Department of Physical Therapy, Daejeon University, Daejeon 34520, Republic of Korea; qodtn11@theresearcher.co.kr (B.-S.K.); samho15@naver.com (S.-H.P.); 2Division of Pulmonary and Critical Care Medicine, Department of Internal Medicine, Chungnam National University Hospital, Daejeon 35015, Republic of Korea; jss24lkm@hanmail.net (S.-S.J.); nextera@naver.com (S.-D.W.); 3Department of Computer Engineering, Daejeon University, Daejeon 34520, Republic of Korea; hjkim99@dju.kr

**Keywords:** spirometry, chronic obstructive pulmonary disease, clinical trial, mobile application, smart phone

## Abstract

A spirometer is a medical device frequently used clinically for the diagnosis and prediction of lung disease. This study aimed to investigate the clinical usefulness of a hand-held spirometer (The Spirokit), compared with conventional spirometry in patients with chronic obstructive pulmonary disease (COPD). This study was conducted from February 2022 to October 2022. Measurements from 80 patients with COPD (male: 53, female: 27) were obtained using The Spirokit and PC-based pulmonary function test equipment, and the resulting values were compared and analyzed. For the concurrent validity comparison of The Spirokit, the intra-class correlation (ICC 2, 1), coefficients of variation (CV_ME_), 95% limits of agreement (95% LOA), and Cohen’s Kappa Index were analyzed. The Spirokit showed high agreement (ICC: 0.929–0.989; 95% LOA: −0.525 to 2.559; and CV_ME_: 0.05–0.08) with the PC-based pulmonary function tester. Using the Cohen’s kappa coefficients, the device showed high sensitivity, specificity, and accuracy scores of Pa: 0.90, Pc: 0.52, and K: 0.79, respectively, indicating considerable agreement. The Spirokit, a portable pulmonary function test device, is a piece of equipment with high validity and portability, with high potential for replacing PC-based pulmonary function test equipment.

## 1. Introduction

The aging rate in South Korea ranks first among the countries of the Organization for Economic Co-operation and Development (OECD). By 2026, it is expected that approximately 20% of the population will be 65 years or older, making the country an ultra-aging society. Considering this trend, South Korea is expected to rank third among the OECD countries in 2050 in terms of the proportion of the older adult population [1]. As life expectancy increases owing to the advancement of modern medicine, most older adults suffer from chronic diseases, wherein chronic obstructive pulmonary disease (COPD) emerges as a prominent concern [2].

According to a 2020 survey by the World Health Organization (WHO), COPD accounts for the third largest cause of death among the global population. Its prevalence in South Korea is also as high as 1 in every 10 adults over 40 years of age [3]. The symptoms of COPD include sputum, pectoralgia, wheezing, cough, and dyspnea. Recognizing the initial symptoms of COPD and its fluctuating nature, which may worsen or improve over time, makes it challenging to detect COPD early. COPD is a chronic disease that continues to worsen, and there is no specific treatment or method to completely cure it. Therefore, continuous management is necessary to prevent symptoms from worsening. A COPD attack is characterized by sudden difficulty breathing, frequent coughing and phlegm, and chest pain. After going to the emergency room, the symptoms subside or the patient is hospitalized for long-term treatment. COPD patients sometimes have seizures even without special circumstances, and such seizures can significantly lower the quality of life of COPD patients. This lowers the early diagnosis rate of COPD to 2.8% in South Korea. COPD has a higher rate of severe illness than other diseases, leading to a higher medical cost [4,5]. COPD not only affects patients’ individual health, but also affects social and economic aspects. COPD is a chronic disease with more than five times the medical costs of diabetes. The cost of the inhaler treatment used is high, and in the case of severe patients, oxygen treatment is required in addition to continuous drug treatment, so the cost of purchasing medical oxygen is continuously incurred. For this reason, not only the patient’s quality of life but also the lives of the patient’s family members can be significantly affected, and it can also have a significant impact on national medical expenses, causing significant losses to the patient, the patient’s family, and the nation [6].

Methods for diagnosing COPD include computed tomography, magnetic resonance imaging, chest X-ray, and pulmonary function tests (PFT). These are used in various ways. Among them, the most commonly used method is the pulmonary function test. The pulmonary function test is a non-invasive test method that can check pulmonary function by considering the characteristics of the air flow rate and flow rate released during breathing and comparing it with the predicted value that matches the subject’s physical condition [7]. The reason pulmonary function tests are most often used to diagnose COPD is because they are non-invasive tests that only test the patient’s flow rate. In the case of other tests, radiation must be used, so although it is a trace amount, radiation must be irradiated, and the exact airway obstruction rate cannot be known [7]. Since a test device is required for the diagnosis, evaluation, and prognosis of both COPD and restrictive ventilation disorder [8], PFTs are used as the primary test for respiratory diseases [9]. Early diagnosis and continuous maintenance are important in COPD, and thus require the frequent use of PFTs. Accordingly, hand-held spirometers with various shapes and functions have been developed [10]. Depending on the configuration, PFTs are divided into all-in-one testers that both test and display the results and detachable testers that test and display the results separately, allowing one to view the results with a smartphone application or PC Software [11]. Portable PFTs connecting to smart devices via Bluetooth have recently been developed. These smart PFTs have no limitations regarding the place and time of testing, and the results are saved in a database, facilitating the understanding of the patient’s health status. Moreover, they enable patients with respiratory diseases to continuously monitor their pulmonary function, helping healthcare providers to determine the medications and the level of treatment, and thus making it useful for clinical decisions [12]. It would be highly beneficial for people with respiratory diseases, as well as clinicians, if hand-held spirometers and breathing training devices that are connected to smart devices were continuously developed [13], with a focus on the precision of the devices and their clinical usefulness. Recently, the development of portable spirometers using smartphones has been actively progressing in Korea. “The Spirokit” (TR Co., Ltd. Daejeon, Republic of Korea) is a wireless-type inspection device that can be inspected through Bluetooth connection with a portable smart device. “The Spirokit” is a wireless portable spirometer that measures flow rate using an infrared turbine propeller. The device is lightweight, and the test software supports Android- and IOS-based operating systems. Therefore, it is designed so that test subjects and examiners can receive test results regardless of the test time and space, and data can be collected through the server and used to calculate lung function prediction formulas. It is an inspection device that helps facilitate inspection by having a guide gauge that can guide the inspection. In previous research, a study investigated the precision of the spirometer through precision measurement along with the development process of “The Spirokit” [14], but evaluations of its clinical usefulness in those with respiratory diseases remains lacking. Thus, this study aimed to investigate the clinical usefulness of spirometers based on the level of agreement with the existing precision PFT device among COPD patients.

The purpose of this study is to verify the validity and clinical usefulness of The Spirokit as a hand-held spirometer by comparing the level of agreement with pulmonary function indicators collected through “The Spirokit” and a precision medical PFT device for COPD patients.

## 2. Materials and Methods

### 2.1. Participants

This study was conducted from February 2022 to October 2022; we recruited 120 COPD outpatients from C University Hospital in D Metropolitan City. The selection criteria were as follows: those who were over 19 years old but under 80 years old and had been diagnosed with COPD by a medical doctor; those who had dyspnea at level 2 or below based on the American Thoracic Society (ATS) [15]; patients with mild symptoms whose measured FEV1% (the ratio of forced expiratory volume in 1 s) was less than 70% and whose predicted value of FEV1 based on ATS was more than 80% [15]; and those who agreed to participate in the study. The exclusion criteria were as follows: those who had visited the emergency room for worsening respiratory symptoms within two months; those who were diagnosed with and treated for acute respiratory diseases (coronavirus disease, acute pneumonia, acute bronchitis, etc.) within 2 months; and those with acute cardiovascular disease. All study participants signed an informed consent form after understanding the objectives and procedures of the study and agreeing to voluntarily participate in the intervention. This study was approved by the bioethics committee of Daejeon University and was registered on the WHO International Clinical Trials Registry Platform (KCT0007736).

### 2.2. Procedures

For this prospective cross-sectional study, we calculated the number of required participants using the equation given by Walter et al. [10]. The number of participants was determined based on a 0.65 acceptable reliability level, a 0.80 expected reliability level, 0.05 statistical significance (α), and 0.80 power (1-β), which resulted in a requirement of at least 77 participants.

To test the validity of the developed hand-held spirometer, “The Spirokit”, its measured PFT variables were compared with those from a precision medical PFT device, “V-Max Encore 22” (Carefusion, San Diego, CA, USA). The test was performed by four clinical pathologists who had at least 5 years of experience in PFTs, rotating randomly by lottery. The procedures were explained to all participants, who underwent a dry run 10 min before the test. In cases where the participants experienced dizziness, dyspnea, pain, or sudden fatigue, the test was immediately stopped to allow them to take a break.

The PFTs were conducted following the guidelines from the ATS. Before conducting a PFT, the participants were informed of the prohibition of drinking for 4 h, eating for 2 h, exercising vigorously for 1 h, smoking for 1 h, and taking beta-agonists, antihistamines, and other drugs that affect the test. In addition, before the PFT, we investigated whether there had been any contraindications to the test, such as ophthalmic surgery, open heart surgery, stroke, heart attack, myocardial infarction, pneumothorax, retinal detachment, or aortic aneurysm, within the past 3 months. And patients who had suffered from a disease that may cause problems with breathing or maximum effort breathing, who were currently infected with tuberculosis or the COVID-19 virus or had been exposed to it, who had had massive hemoptysis within the past month, and whose systolic blood pressure was over 200 mmHg or whose diastolic blood pressure was perceived to exceed 140 mmHg were excluded. If the participants had undergone the above-mentioned surgery or disease, a PFT was not performed. Participants were seated on a chair as instructed by the examiner, with their backs erect and facing 15° forward. The legs were separated by shoulder width, and the hands were placed on the knees. Using a fitted disposable nasal plug, the participants held a disposable inline filter in their mouths, and the examiner ensured that there was no air leakage from the corner of the mouth before the test. The examiner first demonstrated and subsequently instructed the participants to take measurements three times per device. A rest period of 30 min was observed between the measurements. A specialist examined the three measurements to check for any abnormal results, and subsequently collected the highest value. For each device, data were collected from participants in the same posture and method. In cases of errors or mistakes in posture and collection, the measurements were performed again. The forced vital capacity (FVC), forced expiratory volume 1 s (FEV1), and peak expiratory flow (PEF) values, which are clinically important variables in PFTs, were collected and compared [16].

### 2.3. Measurement Methods

#### 2.3.1. Precision Medical PFT Device

The “VMax Encore 22”, based on the principle of a wire sensor, is a mounting-type PFT device that detects the movement of a sensor via a cable and sends the reading to a PC (Figure 1). This device flows a current through the wire, thereby heating it. When the wire cools down due to breathing, it flows a higher current to retain the same temperature and maintain a constant temperature, and the current value is converted to the volume and flow rate for the test.

The dimensions of the test device are 950 mm (high) × 330 mm (wide) × 360 mm (long), weighing 5790 g. The test device has a maximum measurement volume of 12 L and a precision of ±3% for the data. It can measure vital capacity (VC), inspiratory reserve volume (IRV), tidal volume (TV), expiratory reserve volume (ERV), expiratory vital capacity (EVC), inspiratory vital capacity (IVC), forced vital capacity (FVC), forced expiratory volume 1 s (FEV1), peak expiratory flow (PEF), forced expiratory flow between 25% and 75% of functional vital capacity (FEF 25–75%), forced inspiratory vital capacity (FIVC), and maximal voluntary ventilation (MVV). The test results were collected by sending the test reports to the electronic medical record device via wired communication [17].

#### 2.3.2. Hand-Held Spirometer

“The Spirokit”, a hand-held spirometer, is a small test device that is wirelessly connected to a smart device via Bluetooth (Figure 2). The Spirokit calculates flow rates and volumes through a formula that converts the rotation speed of the propellers into a volume through an infrared ray (IR) sensor and a photodetector (PD) sensor. This sensor is an inexpensive sensor and can reduce costs in making devices. It measures flow rates and volumes through a function to compensate for propeller inertia based on flow rate. It is a PFT device approved by the Ministry of Food and Drug Safety of South Korea, and its dimensions are 160 mm (high) × 33 mm (wide) × 50 mm (long), weighing 100 g. Its maximum measurement volume is 12 L and it has ±3% data precision. This device can measure VC, IRV, TV, ERV, RV, IC, EVC, IVC, FVC, FEV1, PEF, FEF 25–75%, and FIVC [18]. The Spirokit’s pulmonary function test software is intended to facilitate patient education, thereby effectively reducing patient examination time. In addition, the test data can be stored on a cloud-based server. the Test results can be viewed on the web, and the server data can be downloaded as CSV-format files.

### 2.4. Statistical Analysis

The mean and standard deviation were calculated from the final data from the precision medical PFT device, VMax Encore 22(Carefusion, San Diego, CA, USA), and hand-held spirometer (The Spirokit). Any differences between the devices were analyzed using a paired *t*-test to examine statistical significance.

To analyze the validity of the the Spirokit, the intraclass correlation coefficient (ICC 2, 1) was applied. An ICC < 0.750 was considered a medium level of agreement, while ICC values between 0.750 and 0.900 and those >0.900 were defined as a good and high levels of agreement, respectively [19].

For an absolute comparison of PFT, the difference between the Spirokit and VMax Encore 22, coefficients of variation of method errors (CV_ME_), and 95% limits of agreements (95% LOA) were calculated [20]. As for the CV_ME_ data, the variation coefficients were calculated using standard deviations from the data of each device and subsequently converted to percentages (ME = Sd/√2, CV_ME_ = 2ME/(X1 + X2) × 100%).

Data were entered and calculated using Microsoft Excel ver. 2022 (Microsoft, Washington, DC, USA). The 95% LOA, Bland–Altman graphs, and scatter graphs were drawn using Medcalc ver. 22.02 (Medcalc software, Osted, Belgium). The Windows SPSS program ver 25.0 (IBM, Armork, NY, USA) was used for the statistical significance analysis of the measured values, and the statistical analysis of the level of agreement between the devices, through which the homogeneity of the measured values between the devices was analyzed. All statistical significance levels (α) were set at 0.05.

## 3. Results

Of the 120 recruited participants, a total of 80 participants were selected after excluding those who refused to participate (*n* = 10), had deterioration of their respiratory symptoms (*n* = 8), and had been taking treatments for acute respiratory diseases (*n* = 22). Fifty-three participants were male and twenty-seven were female. The participants had a mean age of 61.90 years, a mean height of 162.15 cm, a mean weight of 66 kg, and a mean body-mass index of 25.12 kg/m^2^ (Table 1).

The differences in the PFT results between the PPFT device and the Spirokit were analyzed using paired *t*-test, which resulted in significance levels (*p*) of 0.431 for FVC, 0.328 for FEV1, and 0.543 for PEF, indicating no significant differences between the measured values (Table 2).

Regarding the correlation coefficients between the two devices, the ICC values were 0.982 (0.972–0.988) for FVC, 0.989 (0.984–0.994) for FEV1, and 0.929 (0.890–0.955) for PEF, indicating a high level of agreement. The CV_ME_ % values of each variable were as low as 0.05 for FVC, 0.05 for FEV1, and 0.08% for PEF. The 95% LOA values were −0.525 to 0.469 for FVC, −0.271 to 0.355 for FEV1, and −0.321 to 2.559 for PEF, mostly indicating symmetrical distributions (Table 2 and Figure 3).

The Spirokit had 26/33 (78.8%) relative sensitivity, 46/47 (97.9%) specificity, and 72/80 (90%) test accuracy, and the final indices were Pa 0.90, Pc 0.52, and K 0.79, showing a high level of agreement (Table 3).

## 4. Discussion

“The Spirokit” is a turbine propeller-type smart device-based hand-held spirometer developed domestically that can be manufactured in an inexpensive manner. Additionally, because it is an inspection device developed based on an application, it is easy to record and store inspection data. There is also the advantage of being able to conduct tests without restrictions on space and time. In the case of a server-based examination system, a web inquiry system is supported so that patients can be managed using only the Internet through the cloud, and big data can be built by accumulating data. By implementing machine learning based on the inspection data with the constructed big data, the data can be reflected in real time and used for predictive analysis. It also helps in reading the inspection graph, automatically analyzing the graph to determine whether the inspection was performed incorrectly or the inspection results are incorrect. Support is available to easily conduct tests by distinguishing between restrictive, obstructive, mixed ventilatory disorder, and normal ventilation [18].

To assess the comparability of “The Spirokit” against the precision medical PFT device VMax Encore 22, commonly used in hospitals, this study investigated their validities for PFTs in 80 patients with COPD.

Specifically, in this study, we compared the FVC, FEV1, and PEF using the ICC between “The Spirokit” and the existing precision PFT device, which resulted in values of 0.982 for FVC, 0.989 for FEV1, and 0.929 for PEF, indicating a high level of agreement. The 95% LOA determines whether the measured values for each device are within the 95% confidence interval after marking the data on the graph. The result data had, symmetry as shown by the values of −0.525 to 0.469 for FVC, −0.271 to 0.355 for FEV1, and −0.321 to 2.559 for PEF. Moreover, the 95% LOA summarizes the level of disagreement between the two test methods after repeated measurements, in which the level of variation between the measured values was checked to identify the level of agreement between the measured values.

CV_ME_% is a variable for evaluating the level of disagreement among repeated measurements using different tools, which is used to support ICC, as ICC is unable to show the variation among the measured data. A CV_ME_% value <5% is considered a significantly low difference. In this study, there were substantially low variations, as shown by the values of 0.05% for FVC, 0.05% for FEV1, and 0.08% for PEF [19]. These results were consistent with those of Aardal et al. [21], where the validity was compared between a portable ultrasound version of a respiratory disease test device and “VMax Encore 22”, and the authors found a high level of agreement. Moreover, it could be interpreted that “The Spirokit” can generate measurement data that are similar to those of “VMax Encore 22” [22]. However, in contrast to the FVC and FEV1 values, PEF showed a lower validity value. This disparity may be attributed to the design differences between the “VMax Encore 22” and “The Spirokit”. “The VMax Encore 22”, being a wire sensor without the need for inertia compensation, exhibits more sensitive measurement of flow rate compared to “The Spirokit”, which relies on a turbine propeller and necessitates compensation for inertia due to its sensor nature. Despite its low value, the statistical analysis showed a high level of agreement, with a >0.900 ICC, even though it was accompanied by a low coefficient of variation of 0.08%. In addition, most of the data of the 95% LOA were within the threshold, suggesting that “The Spirokit” could be utilized as a clinical device to measure PEF.

This study has certain limitations. First, most recruited patients with COPD were older adults aged 60–70 years. Second, the recruited participants included mild patients with an FEV1% of more than 70%, making it difficult to generalize the results to those with moderate to severe COPD. Finally, we did not study the reliability between the test and retest with “The Spirokit”, and thus, the reliability of the test and retest using this device cannot be guaranteed. The Spirokit has clinical significance as a portable PFT device, providing easy access for patients with limited mobility and those residing in remote areas.

A follow-up study addressing the limitations of this study should be conducted. First, future studies should investigate the applicability of “The Spirokit” for a wide range of patients through a validity study on patients with COPD and asthma under 60 years of age. Second, a validity analysis should be performed on patients with mild to severe COPD to investigate the usefulness of the device in these populations. Third, this study was conducted at a single institution, resulting in a lack of diversity in research sample collection. Therefore, there is a need to conduct additional research at multiple institutions. Last, the reliability between the test and retest should be compared to investigate the homogeneity in the test results of “The Spirokit”.

The following future studies are suggested. “The Spirokit” holds the advantage of being assessable through server data testing, enabling the collection of substantial data with a device capable of handling big data. This facilitates the development of a prediction equation for individuals with normal pulmonary function. Furthermore, it is possible to create pulmonary function prediction equations tailored to normal individuals for each country and region by segmenting test regions into specific zones. For countries or regions lacking prediction equations for normal individuals, specific pulmonary function prediction equations can be developed based on their unique geographical or demographic characteristics. In the future, it is necessary to develop a prediction equation for normal individuals across each country and region using data collected by “The Spirokit”, and subsequently investigate the validity of the equation [23]. Second, collected data from patients can be used for prediction analysis. “The Spirokit” is a home device that can predict the pattern of FEV1 through a time-series analysis based on real-time data of patients. This device would predict the worsening of COPD based on FEV1 patterns, and such variables would help identify future risks. Nonetheless, further studies that compare the reliability of prediction values are warranted.

## 5. Conclusions

This study was conducted to confirm the clinical usefulness of the smart device-based hand-held spirometer “The Spirokit” by demonstrating its concurrent validity. As a result, “The Spirokit” showed sufficient agreement when compared to a medical-precision PFT device. “The Spirokit” can be effectively utilized as an alternative to medical-grade spirometers for screening COPD patients, providing clinical significance as a portable PFT device and easy access to patients with limited mobility and those residing in remote areas.

## Figures and Tables

**Figure 1 healthcare-12-00507-f001:**
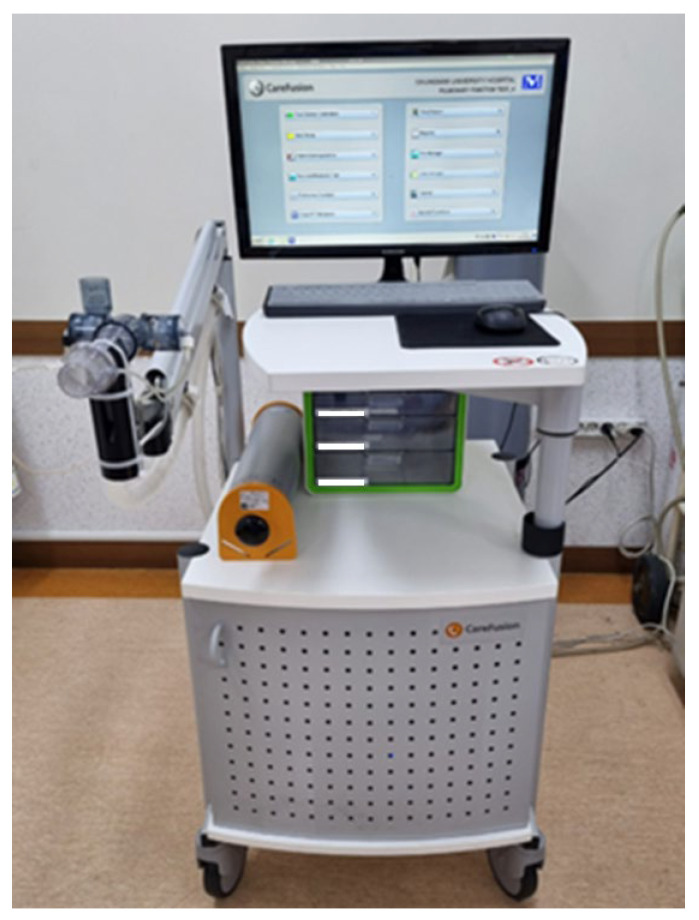
V-Max Encore 22.

**Figure 2 healthcare-12-00507-f002:**
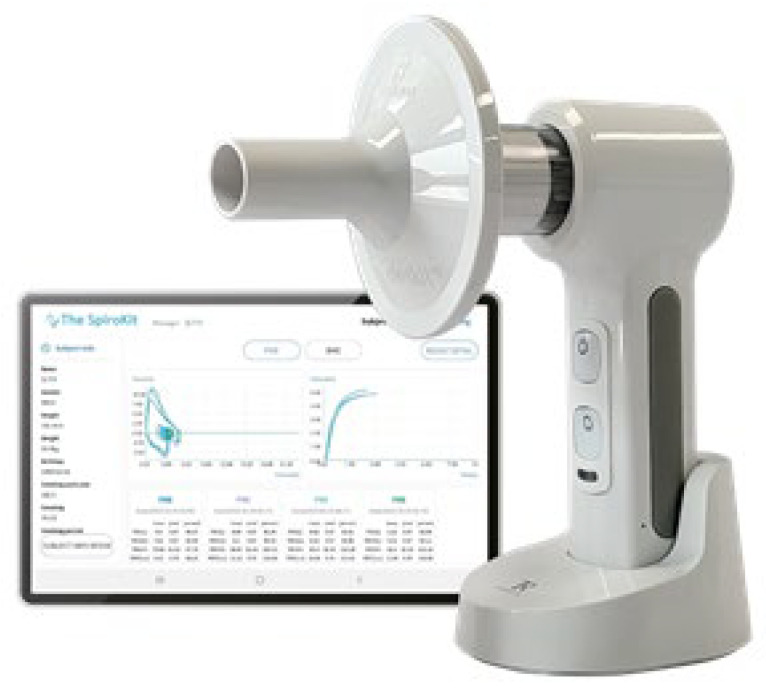
The Spirokit.

**Figure 3 healthcare-12-00507-f003:**
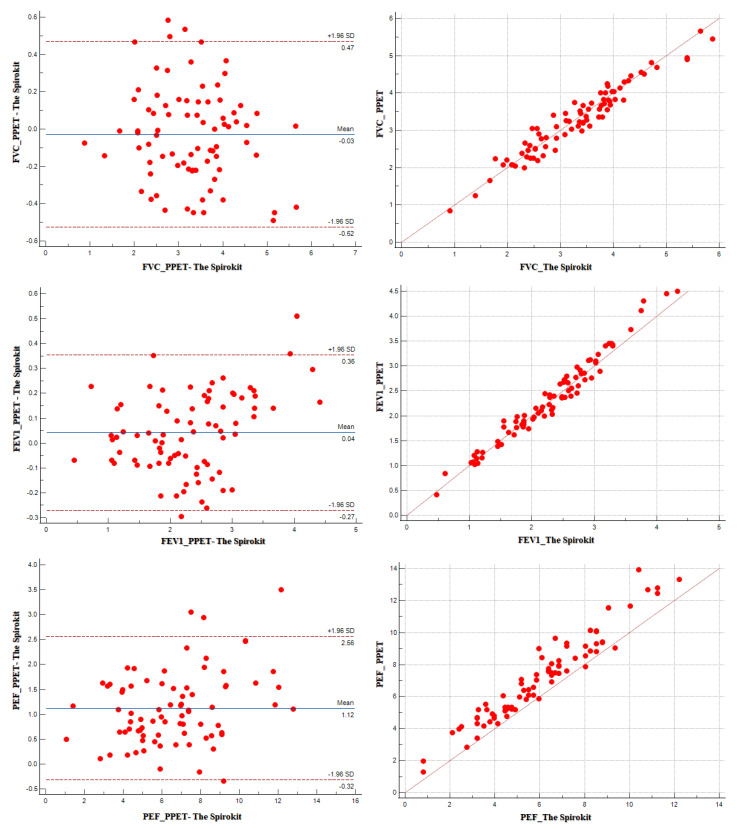
Bland–Atman plot and regression graph of the pulmonary function test variables of the participants between PPFT and the Spirokit.

**Table 1 healthcare-12-00507-t001:** General characteristics of the participants (N = 80).

Variables	Mean ± SD
Sex (male/female)	53/27
Age (years)	61.90 ± 14.02
Height (cm)	162.15 ± 8.51
Weight (kg)	66.00 ± 10.10
BMI (kg/m^2^)	25.12 ± 3.51

Values are presented as mean ± SD. BMI: body mass index.

**Table 2 healthcare-12-00507-t002:** Level of agreement of the pulmonary function test variables for participants using the PPFT and the Spirokit.

Variables	PPFT	The Spirokit	ICC (95% CI)	CV_ME_%	95% LOA	*t* (*p*)
FVC (L)	3.28 ± 0.92	3.30 ± 0.96	0.982 (0.972~0.988)	0.05	−0.525~0.469	1.356 (0.431)
FEV_1_ (L)	2.32 ± 0.84	2.29 ± 0.79	0.989 (0.984~0.994)	0.05	−0.271~−0.355	−0.958 (0.328)
PEF (L/s)	7.19 ± 2.62	6.07 ± 2.40	0.929 (0.890~0.955)	0.08	−0.321~2.559	13.620 (0.543)

Values are presented as mean ± SD. PPTF: PC-based pulmonary function test equipment, FVC: forced vital capacity, FEV_1_: forced expiratory volume in 1 s, PEF: peak expiratory flow, ICC: intra correlation coefficient, CV_ME_%: coefficients of variation of method error %, 95% LOA: 95% limits of agreements.

**Table 3 healthcare-12-00507-t003:** Contingency table of the number of participants diagnosed with obstruction by PC-based spirometer and the Spirokit.

Samples	The Spirokit YES (n)	The Spirokit NO (n)	The Spirokit Total (n)
PPFT YES (n)	26	1	26
PPFT NO (n)	7	46	53
PPFT Total (n)	33	47	80
Calculation coefficients	Pa	Pc	K
	0.90	0.52	0.79

PPTF: PC-based pulmonary function test equipment, Pa: probability of agreement between two raters, Pc: the rate at which two raters coincidentally received a congruent evaluation, K: Cohen’s Kappa coefficients.

## Data Availability

Data are contained within the article.

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
