# Peer review of "Validity Study for Clinical Use of Hand-Held Spirometer in Patients with Chronic Obstructive Pulmonary Disease"

_healthcare, 2024, doi:10.3390/healthcare12050507_

Round 1

Reviewer 1 Report

Comments and Suggestions for Authors

Thank you for the opportunity to review this manuscript on relevant topic. This portable equipment has the potential to reach patients with reduced mobility and populations living in remote areas.

Please find my suggestions and questions.

-Please, enter the date of study in Abstract and Materials and Methods sections.

-Line 106: Please describe “FEV1%”.

-Lines 121-124: “Of the 120 recruited participants, a total of 80 participants were selected after excluding those who refused to participate (n = 10), had deterioration of respiratory symptoms (n = 8), and had been taking treatments for acute respiratory diseases (n = 22).” This information and Figure 1 could be described in Results section.

-Line 129: “The procedures were explained to all participants who had a dry-run before the test” How long to have a dry-run before the test?

Lines 134-135: “…taking drugs that affect the test” Please describe the drugs.

Lines 135-136: “If the test subject violates the prohibitions before the test, test was didn’t performed.” Please correct the phrase.

Lines 139-144: “And whether you suffer from a disease that may cause problems with breathing or maximum effort breathing, whether you are currently infected with tuberculosis or the Covid-19 virus, whether you have been exposed to it, whether you have had massive hemoptysis within the past month, and whether your systolic blood pressure is over 200 mmHg or your diastolic blood pressure is Perception exceeding 140 mmHg was investigated.”

Please correct the phrases.  

Was it investigated or excluded?

Lines 172-173: “It can measure VC, IRV, TV, ERV, RV, IC, EVC, IVC, 172 FVC, FEV1, PEF, FEF25-75%, FIVC, and MVV”.  Please describe.

Does the test performed with “The Spirokit” take the same time as the test performed with the reference device? The authors could include this information in the manuscript.

Lines 215-217: “As for the precision PFT device used as a reference device in this study, the 215 mean values of variables were 3.27L for FVC, 2.32L for FEV1, 70.72% for FEV1/FVC%, and 216 7.18L/s for PEF (Table 1).” Are these mean values general reference values or participant values? If they are general reference values, they should not be described in Table 1. If they are participating values, why are they different from the values described in Table 2?

Discussion

-In the first paragraph, the authors could summarize their work, highlighting the positive results.

- The authors could have commented on the advantage of this portable equipment being able to reach patients with reduced mobility and populations living in remote areas, comparing with the positive results of other portable devices.

Conclusion

-"The Spirokit" can be fully utilized as an alternative to medical precision PFT device to diagnose COPD patients.” Considering the limitations of the study described by the authors, it can be stated that "The Spirokit" can be fully utilized as an alternative to medical precision PFT device to diagnose COPD patients? 

-The authors could highlight the potential of this portable equipment to reach patients with reduced mobility and populations living in remote areas.

Author Response

Reviewer 1

#1 Please, enter the date of study in Abstract and Materials and Methods sections.

Response: Thank you for suggesting. As you suggested, we attached the Abstract and Materials and Methods sections.

“This study was conducted from February 2022 to October 2022.”

#2 Line 106: Please describe “FEV1%”.

Response: Thank you for suggesting. As you suggested, we have revised the main text.

Patients with mild symptoms whose measured FEV1%(the ratio of forced expiratory volume in 1 second) is less than 70% and the predicted value of FEV1 based on ATS is more than 80%

#3 Lines 121-124: “Of the 120 recruited participants, a total of 80 participants were selected after excluding those who refused to participate (n = 10), had deterioration of respiratory symptoms (n = 8), and had been taking treatments for acute respiratory diseases (n = 22).” This information and Figure 1 could be described in Results section.

 Response: Thank you for suggesting. As you suggested, we have revised the main text.

#3 Line 129: “The procedures were explained to all participants who had a dry-run before the test” How long to have a dry-run before the test?

Response: Thank you for your review. we conducted it for 30 minutes prior to the experiment.

“The procedures were explained to all participants who underwent a dry run 10 minutes before the test.”

#4 Lines 134-135: “…taking drugs that affect the test” Please describe the drugs.

Response: Thank you for your review. It refers to medications such as beta-agonists, antihistamines, and others that affect pulmonary function.

“…taking beta-agonists, antihistamines, and other drugs that affect the test.”

#5 Lines 135-136: “If the test subject violates the prohibitions before the test, test was didn’t performed.” Please correct the phrase.

Response: Thank you for suggesting. As you suggested, we have revised the main text.

#6 Lines 139-144: “And whether you suffer from a disease that may cause problems with breathing or maximum effort breathing, whether you are currently infected with tuberculosis or the Covid-19 virus, whether you have been exposed to it, whether you have had massive hemoptysis within the past month, and whether your systolic blood pressure is over 200 mmHg or your diastolic blood pressure is Perception exceeding 140 mmHg was investigated.”

Please correct the phrases.  

Was it investigated or excluded?

Response: Thank you for suggesting. we have excluded those individuals.

#6 Lines 172-173: “It can measure VC, IRV, TV, ERV, RV, IC, EVC, IVC, 172 FVC, FEV1, PEF, FEF25-75%, FIVC, and MVV”.  Please describe.

Response: Thank you for suggesting. As you suggested, we have revised the main text.

“It can measure vital capacity (VC), inspiratory reserve volume (IRV), tidal volume (TV), expiratory reserve volume (ERV), expiratory vital capacity (EVC), inspiratory vital capacity (IVC), forced vital capacity (FVC), forced expiratory volume 1second (FEV1), peak expiratory flow (PEF), forced expiratory flow between 25% and 75% of functional vital capacity (FEF25-75%), forced inspiratory vital capacity (FIVC), maximal voluntary ventilation (MVV).”

#7 Does the test performed with “The Spirokit” take the same time as the test performed with the reference device? The authors could include this information in the manuscript.

 Response: Thank you for suggesting. As you suggested, we have revised the main text.

“'The Spirokit''s pulmonary function test software is visualized to facilitate patient education, thereby effectively reducing patient examination time.In addition, test data can be stored on a cloud-based server.”

#8 Lines 215-217: “As for the precision PFT device used as a reference device in this study, the 215 mean values of variables were 3.27L for FVC, 2.32L for FEV1, 70.72% for FEV1/FVC%, and 216 7.18L/s for PEF (Table 1).” Are these mean values general reference values or participant values? If they are general reference values, they should not be described in Table 1. If they are participating values, why are they different from the values described in Table 2?

Response: Thank you. Your advice was helpful. We have removed the reference values ​​in Table 1.

#9 -In the first paragraph, the authors could summarize their work, highlighting the positive results.

- The authors could have commented on the advantage of this portable equipment being able to reach patients with reduced mobility and populations living in remote areas, comparing with the positive results of other portable devices.

Response: Thank you for suggesting. As you suggested, we attached the discussion sections.

“The Spirokit has clinical significance as a portable PFT device, providing easy access for patients with limited mobility and those residing in remote areas.”

#10 "The Spirokit" can be fully utilized as an alternative to medical precision PFT device to diagnose COPD patients.” Considering the limitations of the study described by the authors, it can be stated that "The Spirokit" can be fully utilized as an alternative to medical precision PFT device to diagnose COPD patients? 

-The authors could highlight the potential of this portable equipment to reach patients with reduced mobility and populations living in remote areas.

Response: Thank you for suggesting. As you suggested, we attached the conclusion sections.

"The Spirokit can be sufficiently utilized as an alternative to medical-grade spirometers for screening COPD patients, providing clinical significance as a portable PFT device, easily accessible to patients with limited mobility and those residing in remote areas."

Reviewer 2 Report

Comments and Suggestions for Authors

This study investigated the clinical usefulness of a hand-held spirometer called "The Spirokit" compared to conventional spirometry in patients with COPD. The objective of the study was to assess the degree of concurrence between the Spirokit and PC-based pulmonary function test equipment.

Overall, the study appears to have several strengths. It includes a decent sample size of 80 patients with COPD, providing a sufficient number of data points for analysis. The study uses various statistical measures such as intra-class correlation, coefficients of variation, and limits of agreement to assess the agreement between the Spirokit and the PC-based equipment. The results show a high level of agreement, indicating that the Spirokit is a valid alternative to PC-based equipment. However, there are some potential concerns and limitations in the study that should be addressed.

1.     The study does not provide details regarding the sampling process for the participants. It is important to ensure that the sample is representative of the target population and that there is no bias in participant selection.

2.     The authors have done an excellent job of furnishing comprehensive information about the distinct features of the Spirokit device. The inclusion of specifics like measurement range, accuracy, and precision is crucial in evaluating the clinical utility of the device. Without these details, it becomes challenging to assess the reliability and validity of the Spirokit as a diagnostic tool for COPD.

3.     What about patient comfort and usability? Could this study address patient comfort or satisfaction with using the Spirokit? Patient experience and ease of use are crucial factors that can affect the practicality and acceptance of a hand-held spirometer in a clinical setting.

4.     Also, the study does not provide information about the experience or training of the individuals who performed the spirometry tests using the Spirokit. Operator proficiency and consistency are crucial factors that can affect the accuracy and reliability of spirometry measurements. It would be helpful to know if the operators were trained and certified in spirometry testing.

5.     Authors may need to elaborate more on the sample characteristics. Details regarding the severity and stage of COPD in the included patients would provide a better understanding of the applicability of the Spirokit in different patient populations. Additionally, information about any comorbidities or concurrent treatments of the participants would be valuable in assessing the generalizability of the study findings.

6.     Should the reference instruments be the most accurate and comprehensive measure of the construct being measured in this study? This should be thoroughly discussed in the study background.

7.     During the PFTs, participants were seated on a chair as instructed by the examiner, with their backs erect and facing 15° forward. I am intrigued to comprehend the reasoning behind the forward tilt of the trunk.

8.     What type of ICCs used to assess reliability? the one-way random effects model ICC, the two-way random effects model ICC, or the two-way mixed effects model ICC.

9.     While the ICC is a commonly used statistic for assessing reliability, it does have some limitations and potential disadvantages. For example, sensitivity to sample characteristics; if the sample is not representative of the JIA, the ICC estimates may not generalize well. Also, the inability to distinguish systematic error: The ICC primarily assesses the relative consistency or agreement among measurements but does not explicitly differentiate between systematic error and random error. It does not provide information about the sources or nature of measurement errors. Therefore, ICC alone may not be sufficient for identifying and addressing specific sources of measurement error.

10.  I would also suggest commenting on the cost-effectiveness or practicality of using the Spirokit compared to the PC-based equipment. It would be beneficial to evaluate the cost, maintenance, and ease of use of the Spirokit in a clinical setting to determine its feasibility for widespread adoption.

11.  The study was conducted at a single center in South Korea. The results may not be generalizable to other populations or healthcare settings with different patient demographics, equipment standards, or clinical practices.

12.  It is worth expanding on the implications of the findings to practice in the “discussion” section. This is currently lacking.

Author Response

#1 The study does not provide details regarding the sampling process for the participants. It is important to ensure that the sample is representative of the target population and that there is no bias in participant selection.

Response: Thank you for your advice. We are attaching the content regarding the sampling process in procedure session.

“The number of participants was determined based on 0.65 in acceptable reliability level, 0.80 in expected reliability level, 0.05 in statistical significance (α), and 0.80 in Power (1-β), which resulted in a requirement of at least 77 participants.”

#2 The authors have done an excellent job of furnishing comprehensive information about the distinct features of the Spirokit device. The inclusion of specifics like measurement range, accuracy, and precision is crucial in evaluating the clinical utility of the device. Without these details, it becomes challenging to assess the reliability and validity of the Spirokit as a diagnostic tool for COPD.

Response: Thank you for your advice. We are attaching the relevant content in Measurement Methods session.

“It is a PFT device approved by the Ministry of Food and Drug Safety of South Korea, and its di-mensions are 160 mm (high) × 33 mm (wide) × 50 mm (long), weighing 100 g. Its maxi-mum measurement volume is 12 L and has ±3% data precision. This device can measure VC, IRV, TV, ERV, RV, IC, EVC, IVC, FVC, FEV1, PEF, FEF25-75%, and FIVC [18].”

#3 What about patient comfort and usability? Could this study address patient comfort or satisfaction with using the Spirokit? Patient experience and ease of use are crucial factors that can affect the practicality and acceptance of a hand-held spirometer in a clinical setting.

Response: Thank you for your review. This study is designed to determine whether the hardware and software of 'The Spirokit' can replace existing inspection devices. All participants in this study were satisfied with the test process, and the high validity shown in the research results proves that this test is convenient enough to replace existing tests.

#4 Also, the study does not provide information about the experience or training of the individuals who performed the spirometry tests using the Spirokit. Operator proficiency and consistency are crucial factors that can affect the accuracy and reliability of spirometry measurements. It would be helpful to know if the operators were trained and certified in spirometry testing.

Response: Thank you for your review. The experimental group may be familiar with existing pulmonary function tests. However, I can say that I have no experience with testing using this device, as this is my first experience with the testing provided by 'The Spirokit'. The high validity shown in the research results proves that 'The Spirokit' test software is convenient enough to replace existing tests.

#5 Authors may need to elaborate more on the sample characteristics. Details regarding the severity and stage of COPD in the included patients would provide a better understanding of the applicability of the Spirokit in different patient populations. Additionally, information about any comorbidities or concurrent treatments of the participants would be valuable in assessing the generalizability of the study findings.

Response: Thank you for your review. The diagnostic criteria is that the test was conducted on mild patients who have difficulty breathing below level 2 of the ATS standard.

“Selection criteria were as follows: those who are over 19 years old but under 80 years old and have been diagnosed with COPD by a medical doctor; and those who had dyspnea at level 2 or below based on the American Thoracic Society (ATS) [15]; Persons with mild symp-toms who are 70% or more of the measured FEV1% (FVC divided by FEV1, expressed as a percentage) value and predicted FEV1% value based on ATS[15]; Those who agreed to participate in the study.”

“The exclusion criteria were as follows: those who visited the emergency room for worsening respiratory symptoms within two months, and those who were diagnosed and treated for acute respiratory diseases (coronavirus disease, acute pneumonia, acute bronchitis, etc.) within 2 months; Those with acute cardiovascular dis-ease. All study participants signed the informed consent form after understanding the ob-jectives and procedures of the study, and agreeing to voluntarily participate in the inter-vention.”

#6 Should the reference instruments be the most accurate and comprehensive measure of the construct being measured in this study? This should be thoroughly discussed in the study background.

Response: Thank you for your review. The pulmonary function equipment that is currently most commonly used in Korean university hospitals and has academic references was selected as the equipment to be compared.

“The “VMax Encore 22”, based on the principle of wire sensor, is a mounting type PFT device that detects the movement of the sensor via cable and sends it to the PC (Figure 2). This device flows current through the wire thereby heating it. When the wire cools down due to breathing, it flows a higher current to retain the same temperature and maintain a constant temperature, in which the current value is converted to volume and flow rate for the test. The dimensions of the test device are 950 mm (high) × 330 mm (wide) × 360 mm (long), weighing 5,790 g. The test device has a maximum measurement volume of 12 L and a precision of ±3% for the data.”

#7 During the PFTs, participants were seated on a chair as instructed by the examiner, with their backs erect and facing 15° forward. I am intrigued to comprehend the reasoning behind the forward tilt of the trunk.

Response: Thank you for your review. The pulmonary function tests in this study were conducted according to the pulmonary function testing guidelines provided by the ATS.

#8 What type of ICCs used to assess reliability? the one-way random effects model ICC, the two-way random effects model ICC, or the two-way mixed effects model ICC.

Response: Thank you for the review. ICC [2.1], a two-way random effects model ICC, was used. ICC [2,1] is the same as two-way random effects model ICC.

#9 While the ICC is a commonly used statistic for assessing reliability, it does have some limitations and potential disadvantages. For example, sensitivity to sample characteristics; if the sample is not representative of the JIA, the ICC estimates may not generalize well. Also, the inability to distinguish systematic error: The ICC primarily assesses the relative consistency or agreement among measurements but does not explicitly differentiate between systematic error and random error. It does not provide information about the sources or nature of measurement errors. Therefore, ICC alone may not be sufficient for identifying and addressing specific sources of measurement error.

Response: Thank you for your review. As your review, it is difficult to prove all validity with ICC [2,1]. Therefore, the CVME% value, 95%LOA, and the corresponding Bland-Altman plot are indicated. This value is explained in section 2.4 Statistical Analysis (lines 198-218).

#10 I would also suggest commenting on the cost-effectiveness or practicality of using the Spirokit compared to the PC-based equipment. It would be beneficial to evaluate the cost, maintenance, and ease of use of the Spirokit in a clinical setting to determine its feasibility for widespread adoption.

Response: Thank you for your review. The convenience of 'The Spirokit' is additionally described in Method 2.3.2(lines 183-197).

#11 The study was conducted at a single center in South Korea. The results may not be generalizable to other populations or healthcare settings with different patient demographics, equipment standards, or clinical practices.

Response: Thank you for your advice. We are attaching the relevant content for the main text.

“The study was conducted at a single institution, resulting in a lack of diversity in research sample collection. Therefore, there is a need to conduct additional research at multiple institutions.”

#12 It is worth expanding on the implications of the findings to practice in the “discussion” section. This is currently lacking.

Response: Thank you for suggesting. As you suggested, we attached the discussion sections.

“The Spirokit has clinical significance as a portable PFT device, providing easy access for patients with limited mobility and those residing in remote areas.”

Round 2

Reviewer 2 Report

Comments and Suggestions for Authors

The authors have effectively responded to the previous comments and suggestions. In my opinion, the article is now prepared for publication in its present state.